# Oxymatrine Improves Oxidative Stress-Induced Senescence in HT22 Cells and Mice via the Activation of AMP-Activated Protein Kinase

**DOI:** 10.3390/antiox12122078

**Published:** 2023-12-06

**Authors:** Nagarajan Maharajan, Chang-Min Lee, Karthikeyan A. Vijayakumar, Gwang-Won Cho

**Affiliations:** 1Department of Biological Science, College of Natural Sciences, Chosun University, 309 Pilmun-daero, Dong-gu, Gwangju 501759, Republic of Korea; geneticnaga1990@gmail.com (N.M.); dlckdals274@naver.com (C.-M.L.); cartkn1991@gmail.com (K.A.V.); 2BK21 FOUR Education Research Group for Age-Associated Disorder Control Technology, Department of Integrative Biological Science, Chosun University, Gwangju 61452, Republic of Korea; 3The Basic Science Institute of Chosun University, Chosun University, Gwangju 61452, Republic of Korea

**Keywords:** Oxymatrine, AMPK, cellular senescence, oxidative stress, autophagy

## Abstract

The accumulation of oxidative stress is one of the important factors causing cellular senescence. Oxymatrine (OM) is a natural quinolizidine alkaloid compound known for its antioxidant effects. This study aimed to investigate the anti-senescence potential of OM through oxidative stress-induced in vitro and in vivo models. By treating 600 μM of H_2_O_2_ to the HT22 mouse hippocampal neuronal cell line and by administering 150 mg/kg D-galactose to mice, we generated oxidative stress-induced senescence models. After providing 1, 2, and 4 μg/mL of OM to the HT22 mouse cell line and by administering 50 mg/kg OM to mice, we evaluated the enhancing effects. We evaluated different senescence markers, AMPK activity, and autophagy, along with DCFH-DA detection reaction and behavioral tests. In HT22 cells, OM showed a protective effect. OM, by reducing ROS and increasing p-AMPK expression, could potentially reduce oxidative stress-induced senescence. In the D-Gal-induced senescence mouse model, both the brain and heart tissues recovered AMPK activity, resulting in reduced levels of senescence. In neural tissue, to assess neurological recovery, including anxiety symptoms and exploration, we used a behavioral test. We also found that OM decreased the expression level of receptors for advanced glycation end products (RAGE). In heart tissue, we could observe the restoration of AMPK activity, which also increased the activity of autophagy. The results of our study suggest that OM ameliorates oxidative stress-induced senescence through its antioxidant action by restoring AMPK activity.

## 1. Introduction

During cellular senescence, cells lose their ability to divide; i.e., they enter a cell cycle arrest state [1,2,3]. This condition is harmful but sometimes also plays an important biological role. Its role arrests the cycle of cells with damaged DNA or tumor mutations and maintains tissue homeostasis through repair [4,5]. The accumulation of senescent cells impedes tissue function, reduces regenerative power through inflammatory reactions, and increases the probability of cancer development [6]. It is also a cause of geriatric diseases [7]. The causes of cellular senescence include replicative senescence, during which telomeres shorten, and stress-induced senescence, which occurs in response to stress [8,9]. Reducing cellular senescence is expected to extend one’s lifespan and alleviate geriatric diseases [7]. For this purpose, new materials are being researched and studied. Stress-induced premature senescence (SIPS) is a method used to create senescence models using artificial stress in vitro and in vivo. This method is frequently used to understand the mechanisms of senescence [10,11]. Hydrogen peroxide (H_2_O_2_) and D-galactose (D-Gal) accumulation are examples of SIPS [12,13].

Reactive oxygen species (ROS) are involved in homeostasis maintenance. However, excessive ROS accumulation disrupts homeostasis and causes damage [14]. The excessive accumulation of ROS is called oxidative stress [15]. Oxidative stress disrupts antioxidant function, causing damage to biomolecules and mitochondrial dysfunction [16]. This leads to senescence. This may indicate that oxidative stress is one of the fundamental mechanisms in the senescence process [17,18]. Highly reactive H_2_O_2_ causes oxidative stress while inducing biomolecule damage and apoptosis [19]. The accumulation of D-galactose causes oxidative stress through reducing antioxidants and increasing the levels of ROS [13]. Further, during D-galactose metabolism, non-enzymatic glycosylation reactions lead to the formation of advanced glycation end products (AGEs). The accumulation of AGEs promotes oxidative stress and inflammatory responses and contributes to neurodegenerative diseases [20].

AMP-activated protein kinase (AMPK) is a serine/threonine protein kinase, a heterotrimeric protein complex consisting of three subunits: α, β, and γ [21]. It also activates autophagy for energy recycling. Additionally, AMPK responds to various stresses, including oxidative stress. Therefore, the activation of AMPK enhances cellular stress tolerance and metabolic efficiency [22,23]. This extends lifespans and reduces metabolic disorders [24,25]. The AMPK’s responses to oxidative stress are as follows: (1) to facilitate an increase in antioxidant enzymes by nuclear factor erythroid 2-related factor 2 (Nrf2); (2) to facilitate an increase in autophagy activity, including mitophagy and mitochondrial biogenesis; and (3) to facilitate a reduction in energy surplus through the inhibition of ROS production [26,27]. Ultimately, oxidative stress activates AMPK, as a stress response process. This correlation indicates that AMPK has important functions in homeostasis, senescence, and metabolic disorders [24,25,28].

*Sophora flavescens* is a medicinal plant that has been used in traditional Chinese medicine and also in traditional Korean medicine for a long time. People use its root extensively for medicinal purposes. The benefits of this plant include skin and digestive function enhancement, liver protection, and immunity enhancement [29]. Oxymatrine (OM) is a natural quinolizidine alkaloid compound extracted from the roots of *Sophora flavescens*. Its molecular Formula is C_15_H_24_N_2_O_2_. It is commonly known for its hepatoprotective role and has antioxidant effects because it has a quinolizidine ring structure [30,31,32]. Recent studies have revealed that OM has various functions, including antibacterial, antiviral, anti-inflammatory, and anticancer functions [33,34]. Various studies have also shown its role in improving blood pressure regulation and ischemia/reperfusion injury in the heart and that it has a protective effect on nerves, so it has been used in neurodegeneration models [35,36,37].

We conducted this study to determine whether the antioxidant property of OM affects in anti-senescence. Although, from numerous past studies, we know the various functions of OM, its response to senescence is clearly unknown. Therefore, considering the fact that OM has an ameliorating effect on neurological and cardiac diseases, (1) in vitro, we evaluated the neuroprotective effects of OM using HT22 mouse hippocampal neuron cell lines and the anti-senescence response to a H_2_O_2_-induced senescence model, and (2) in vivo, we evaluated neuronal and cardiac improvements and responses to senescence through antioxidants in a D-Gal-induced senescence model. The results suggest that OM ameliorates oxidative stress-induced senescence through an antioxidant response that restores the activity of AMPK.

## 2. Materials and Methods

### 2.1. Chemicals and Reagents

Sigma-Aldrich (Saint Louis, MO, USA) provided the Methylthiazolyldiphenyl-tetrazolium bromide (#M6494 MTT) assay, 2′,7′-dichlorofluorescin diacetate (#D6883), H_2_O_2_ (#216763), and D-Gal (#G0750). ChemFaces (430056; Wuhan, China) supplied Oxymatrine (#CFN99805; purity ≥ 99.4%). Santa Cruz Biotechnology (Dallas, TX, USA) supplied the Radioimmunoprecipitation (RIPA) lysis buffer, Pierce BCA protein assay kit, and Hoechst 33342 Trihydrochloride Trihydrate (#H3570). Thermo Fisher Scientific (Waltham, MA, USA) provided RNAiso Plus (#9109; Total RNA extraction reagent), and Primescript TM II 1st strand cDNA synthesis kits (#6210A) were purchased from Takara Bio Inc. (Shiga, Japan). Santa Cruz Biotechnology, Inc. (Dallas, TX, USA), supplied the primary antibodies, including p53 (#sc-6243), p21 (#sc-397), p16 (#sc-1661), LC3α/β (#sc-398822), BECN1 (#sc-48341), SQSTM1 (#sc-48402), and GAPDH (#sc-365062). Cell Signaling Technology (Danvers, MA, USA) provided AMPKα (#5832) and p-AMPK (#2535). Santa Cruz Biotechnology, Inc. (Dallas, TX, USA), supplied HRP-conjugated secondary antibodies, including mouse anti-rabbit (#sc-2357) and mouse anti-goat (#sc-2354) antibodies, while Cell Signaling Technology (USA) provided horse anti-mouse antibodies (#7076). GE Healthcare (Buckinghamshire, UK) supplied ECL Western blotting detection reagents (RPN2209) and ECL Select Western blotting detection reagents (RPN2235).

### 2.2. HT22 Cell Culture

We used the high-glucose Dulbecco’s modified Eagle medium (DMEM) containing L-glutamine for the HT22 cell cultures. The media was supplemented with 1% sodium pyruvate solution (Gibco, Life Technologies, New York, NY, USA), 1% penicillin/streptomycin solution (Lonza, Walkersville, MD, USA), and 10% FBS (Gibco, Life Technologies, USA). The cells were incubated at 37 °C with 5% CO_2_ and saturated humidity. We replaced the media every two days. Upon reaching 70% confluence, we subcultured the cells. Once subcultured, the cells were grown in serum-free media for further experiments.

### 2.3. Cell Viability Assay

The methylthiazolyldiphenyl-tetrazolium bromide (MTT) test was performed to measure the toxicity and cytoprotective effects of OM in the HT22 cells. To evaluate the toxicity of the OM, the HT22 cells were cultured for 24 h in a 96-well plate at a density of 2 × 10^3^ per well. Afterwards, the cells were treated with different concentrations of OM (0.5–10 μg/mL) in serum-free culture media for 24 h and incubated with MTT solution for 2 h. H_2_O_2_ was used to induce oxidative stress-induced cellular senescence. To select the appropriate concentration of H_2_O_2_ for the subsequent experiments, cell viability was evaluated by culturing the cells with H_2_O_2_ (100–800 μM) in serum-free culture media for 6 h. Following the 6 h incubation the cells were treated with MTT solution and incubated for 2 h. Formazan crystals were dissolved using dimethyl sulfoxide (DMSO). A total of 600 μM of H_2_O_2_ was chosen for the subsequent experiments.

To evaluate the cytoprotective effect of OM, firstly, the cells were pretreated with multiple concentrations of OM (0.5–8 μg/mL) in serum-free culture media for 24 h and then treated with 600 μM H_2_O_2_ in serum-free culture media. Thereafter, cell viability was evaluated by culturing with MTT solution for 2 h. Cell viability was measured using SpectraMax ABS Plus (Molecular Devices, Thermo Fisher Scientific) at 570 mm wavelength. The experiment was conducted in serum-free culture media, including the control group.

### 2.4. Detection of Intracellular ROS

To measure intracellular ROS, we used the detectable 2′,7′-dichlorofluorescin diacetate (DCFH-DA) fluorescent substance. On a 6-well cell culture plate, we seeded the HT22 cells at a density of 3 × 10^4^ per well for 24 h. OM concentrations (1, 2 and 4 μg/mL) were pretreated in serum-free culture media for 24 h, and 600 μM H_2_O_2_ was post-treated in serum-free culture media for another 6 h. Afterwards, they were incubated with 20 μM DCFH-DA in PBS solution for 30 more minutes. We used Hoechst 33342 in PBS solution for 30 min to stain the cell nuclei. The experiment was conducted in serum-free culture media, including the control group. Then, the cells were observed using a fluorescent microscope (Nikon Eclipse Ti2, Tokyo, Japan), and images were captured (Nikon DS-Ri2, Japan) and analyzed using ImageJ software v1.4.3.x.

### 2.5. Immunoblotting Analysis

We used RIPA buffer containing phenylmethylsulfonyl fluoride (PMSF), sodium orthovanadate (Na3VO4), and a protease inhibitor cocktail (ThermoFisher scientific, Rockford, IL, USA) to extract total protein. After incubation for 30 min at 4 °C, the samples were centrifuged at 16,000× *g* rcf for 20 min. A total of 20–30 μg protein was separated by size through sodium dodecyl sulphate–polyacrylamide gel electrophoresis (SDS-PAGE), and the proteins were blotted on a PVDF membrane (GE Healthcare, Munich, Germany). We the blocked the membranes with 1% blocking solution containing non-fat dry skim milk in TBS-T. We then incubated the membrane overnight at 4 °C with the primary antibody. We used horseradish peroxide-conjugated secondary antibodies and visualized the blots using ECL and pico-ECL (GE Healthcare).

### 2.6. Animals and Administration of Drugs

Male, six-week-old C57BL/6 mice (weighing 22 ± 2 g) were acquired from Samtako sBio Korea Co., Ltd. (Osan, Gyeonggi, Republic of Korea) and kept at 23–25 °C under a 12 h light/dark cycle with ad libitum access to water and food in a sterile facility. The Animal Care and Use committee, of Chosun University sanctioned all the animal experiments (CIACUC2020-A0009). After one week of adaptation, we randomly divided the animals into three groups (4 mice per group): normal control group (PBS alone), D-Gal model group (150 mg/kg/day), and D-Gal + OM (50 mg/kg/day) group. We administered PBS or D-Gal to the mice through intraperitoneal injection for 10 weeks and OM for 8 weeks starting from the 3rd week. Once in every week, we measured the body weight of the animals.

### 2.7. Open Field Test

We evaluated the mice locomotor activity and anxiety using an open field test. We acclimatized the mice in the testing room for 30 min for the open field test. After 30 min of acclimation, we placed the mice in the middle of an open field (length, width, height (cm)). Parameters including the locations, paths, and the time spent in the center square for each animal were recorded for 5 min by using ToxTrac, a freely available software, to track the organisms [38]. We considered the total distance travelled by the mouse as locomotor activity, and movement around the arena’s perimeter was quantified as an anxiety index.

### 2.8. Morris Water Maze Test

We conducted the Morris water maze test at the end of the behavior test to measure the spatial memory ability of the mice in each group. We used a circular water tank (100 cm in diameter, 40 cm in height), divided it into four quadrants, and filled it with water (23 ± 2 °C) to a depth of 15.5 cm. We rendered it opaque by adding non-toxic white paint. We placed a hidden platform at the midpoint of one of the quadrants 1 cm below the water surface. For the training, we introduced each mouse in the maze facing towards the pool wall in quadrants 1, 2, and 3, respectively. During the test, the mice should find the submerged platform in 60s and should stay on it for 30 s. We drove the mice that took longer than 60 s to locate the submerged platform and kept them there for 30 s, and the escape latency was recorded as 60 s. On day 5, we removed the submerged platform to conduct the probe to evaluate memory consolidation. During this, we allowed the mice to swim freely for 60 s, and we measured the time spent by the mice in the target quadrant (where the hidden platform was placed during the training). The time spent on the target quadrant was considered as the degree of memory consolidation. All data were analyzed using ToxTrac Version:2.96, a freely available software, to track the organisms [38].

### 2.9. RNA Extraction and Quantitative Reverse Transcription-Polymerase Chain Reaction (qRT-PCR)

To isolate total RNA from hippocampus and heart tissue, we used RNAisoPlus (Takara). We reverse-transcribed 2.5 μg total RNA using the Primescript II 1st strand cDNA synthesis kit (Takara) and used the Power SYBR Green PCR Master mix (Applied Biosystems, Waltham, MA, USA) for normalization. Mouse primers [the β-actin (NM_007393.5) forward primer 5′-CCACCATGTACCCAGGCATT-3′ and reverse primer 5′-CGGACTCATCGTACTCCTGC-3′, as well as the RAGE (NM_001136.5) forward primer 5′-AGGTGGGGACATGTGTGTC-3′ and reverse primer 5′-TCTCAGGGTGTCTCCTGGTC-3′] were used for amplification. We purchased the primer pairs from GenoTech (Daejeon, Republic of Korea) and IDT (Integrated DNA Technologies, Coralville, IA, USA). We used the StepOne Real-Time PCR system (Applied Biosystems) to conduct the real-time PCR reactions.

### 2.10. Statistical Analyses

Data are represented as the mean ± standard deviation measured from at least three biological replicates. Significance between data sets was evaluated using Student’s *t*-test and an analysis of variance (ANOVA) followed by a post hoc Holm–Sidak’s multiple comparison test using GraphPad Prism Version 8.0.1 (GraphPad Software). Levels of statistical significance are indicated in the figures using asterisks and other symbols (*^/#^ *p* < 0.05, **^/##^ *p* < 0.01, ***^/###^ *p* < 0.001, ****^/#####^ *p* < 0.0001). # is the statistical significance between the senescence model and the OM group.

## 3. Results

### 3.1. Oxymatrine Protects the HT22 Cells against Oxidative Stress

First, the effect of OM on the viability of HT22 cells was evaluated. To do so, we treated the cells with OM at different concentrations, ranging from 0.5 to 10 μg/mL, for 24 h. Upon treatment, there were no significant changes in cell viability (Figure 1A). Additionally, upon treatment with H_2_O_2_, cell viability was also evaluated. Concentrations of 100–800 μM were administered for 6 h. We observed a significant decrease depending on the concentration, but a sharp decline in viability was observed from 700 μM (Figure 1B). To induce oxidative stress, we chose 600 μM, as the viability was close to 60%. Finally, we evaluated the protective effect of OM under oxidative stress caused by H_2_O_2_. OM was pretreated at a concentration of 0.5–8 μg/mL for 24 h and then post-treated with H_2_O_2_ 600 μΜ for 6 h. As a result, we observed that cell viability increased in a dose-dependent manner (Figure 1C). These results suggested that OM has a protective effect against oxidative stress.

### 3.2. OM Reduces ROS Caused by Oxidative Stress in HT22 Cells

We evaluated the effect of OM on oxidative stress-induced ROS generation and accumulation. We found that ROS significantly increased with H_2_O_2_ treatment, but ROS decreased in a dose-dependent manner in the OM pretreatment group (Figure 2A,B). This result suggest that OM can alleviate ROS generation and accumulation caused by oxidative stress.

### 3.3. OM Reduces Oxidative Stress-Induced Senescence via the Activation of AMPK in HT22 Cells

We assessed whether OM treatment restores the activity of AMPK and reduces senescence in a H_2_O_2_-induced senescent cell model. In the oxidative stress-induced group (H_2_O_2_ 600 μΜ), AMPK activity was significantly decreased, while AMPK activity was significantly recovered in a dose-dependent manner (Figure 3A,B). We evaluated the anti-senescence effect of OM through the expression of senescence markers p53, p21, and p16. The expression of senescence markers increased in the H_2_O_2_ group, while it was observed to be significantly decreased in the OM-treated group (Figure 3C–F). This result suggests that the activation of APMK by OM reduces oxidative stress and thus lowers the senescence caused by oxidative stress.

### 3.4. Oxymatrine Improves Behavioral Dysfunction in D-Gal-Induced Senescence Mice

We conducted further experiments using a D-Gal-induced senescence mice model. To make the model, we treated the mice with 150 mg/kg of D-Gal. We administered 50 mg/kg of OM for therapeutic purposes (Figure 4A). Body weight was measured weekly until the end of the experiment. We did not observe any weight differences in any of the groups (Figure 4B). To monitor spatial learning and memory, we conducted the Morris water maze test. First, we measured the time taken by a mouse to identify the platform. We did not notice any differences between the D-Gal and OM groups during the 3-day test period. On the last day, we observed that the OM group took less time to find the platform than the D-Gal group (Figure 4C). The OM group also stayed on the platform longer than the D-Gal group (Figure 4D). Finally, when we removed the platform, the OM group crossed the specific region of the quadrant where the platform was placed previously more often than the D-Gal group (Figure 4E). This result suggests that the administration of OM could restore memory deteriorated by senescence.

### 3.5. Oxymatrine Improved Locomotor Activity and Reduced Anxiety Symptoms in D-Gal-Induced Senescence Mice

Next, we conducted a locomotor activity open field test to observe activity levels, exploration intention, and anxiety symptoms. Figure 5A shows the overall movement of the three groups of mice. We observed a significant decrease in the exploration rate and total distance travelled in the D-Gal group. Meanwhile in the mice treated with OM, we observed a significant increase. (Figure 5B,C). When a mouse is anxious, it becomes less mobile and tends to stay near the walls in the field. To evaluate this phenomenon, we measured the exploration rate and distance from the center of the field. In the D-gal group, we observed a significant decrease in the exploration rate and distance from the center, and the mice stayed at a specific area. On the other hand, we observed increased movement in the OM group (Figure 5D,E). This result suggests that OM participates in neurological recovery and reduces anxiety symptoms.

### 3.6. Oxymatrine Reduces RAGE Expression and Hippocampal Senescence via the Activation of AMPK in D-Galactose-Induced Senescence Mice

The long-term administration of D-Gal can cause neuroinflammation, which can cause problems with cognitive function. This can be associated with signs of brain aging. We evaluated the expression of RAGE (a senescence-related inflammatory marker) at the mRNA level to investigate whether OM administration could reduce hippocampal inflammation. We found an increase in the expression of RAGE in the D-Gal group when compared to the control group, whereas it was significantly decreased in the OM group (Figure 6A). This suggests that OM administration can reduce the expression of RAGE and alleviate hippocampal inflammation. Additionally, in the D-Gal-induced senescence mouse model, we evaluated the anti-senescent effects of OM in the hippocampal tissue. AMPK activity between the D-Gal group and control group did not show any significant changes, but when comparing the D-gal group and the OM group, the AMPK activation in the OM group was significantly increased (Figure 6B,C). Furthermore, we assessed the senescence marker p53. It was also decreased in the OM group (Figure 6D,E). These results suggest that OM could ameliorate senescence through reducing oxidative stress by restoring AMPK activity from the hippocampus.

### 3.7. Oxymatrine Reduces Senescence via the Activation of AMPK and Autophagy in D-Gal-Induced Senescence Mice Heart Tissue

Finally, we investigated the heart tissues obtained from the D-Gal-induced senescence mice treated with OM. AMPK activity in the D-Gal group did not show any significant changes when compared with the control, but we did observe a significant increase in activity upon OM treatment (Figure 7A,B). Furthermore, we found a deterioration of autophagy activity in heart tissue in the D-Gal group, whereas OM administration increased the expression of autophagy markers (Figure 7C–F). Further, we evaluated the senescence markers p53 and p21 in heart tissue. The expression of senescence markers was found to be decreased (Figure 7G–I). This result suggests that OM could restore AMPK activity in cardiac tissues to help autophagy activation and reduce senescence.

## 4. Discussion

Oxymatrine is a quinolizidine alkaloid compound that has been shown in numerous studies to have various beneficiary functions, including antioxidant effects; anti-inflammatory, antiviral, and anticancer activities; immune system regulation; cardiovascular improvement; and nervous system protection [33,34,35,36,37]. It can activate the Nrf2 pathway and enhance antioxidant enzymes [39,40]. Several studies have reported that OM treatment could improve ischemia/reperfusion and liver failure through the Nrf2/ho-1 pathway [30,31,41]. It has also been shown to have effects on neuroprotection, hepatoprotection, and inflammatory responses through the SIRT1 pathway [42,43,44].

Interactions between SIRT1 and AMPK have also been documented [45,46]. This shows that OM can activate AMPK. AMPK regulates the homeostasis of energy metabolism. Therefore, it is referred to as a metabolic master switch. Additionally, AMPK confers adaptive properties to stress [47]. This role of AMPK is associated with senescence [48]. From a metabolic perspective, excessive energy consumption accelerates aging [49]. Supporting this, there are reports stating that calorie restriction slows down aging [50,51,52]. This shows that the regulation of AMPK activity is closely related to energy homeostasis and longevity [23]. Oxidative stress is also one of the main causes of cellular senescence [18]. A study showed that AMPK enhances oxidative stress resistance through its downstream regulators [53]. The Nrf2 pathway also increases the levels of antioxidants [54]. In addition, AMPK also interacts with SIRT1 to regulate cellular energy metabolism, apoptosis, cell proliferation, and inflammatory responses. SIRT1 is known to directly activate autophagy [55]. Ultimately, AMPK regulates metabolic homeostasis and cell survival during metabolic and oxidative stress [56], and it regulates aging-related transcription factors [57]. The activation of AMPK has also been shown to alleviate ischemia/reperfusion symptoms and have neuroprotective effects [58,59]. These functions are associated with the biochemical characteristics of OM. This suggests that OM can activate AMPK. As such, OM is closely related to AMPK and is thought to be able to respond to oxidative stress, as well as reduce cellular senescence.

This study focused on the function of OM in regulating the AMPK pathway and investigated how OM works against senescence. We observed that OM exhibited a protective effect in a cellular senescence model (Figure 1) and that the restoration of AMPK activity reduced senescence (Figure 3). In the case of hippocampal tissue in animal models, behavioral tests showed that OM supported neurological recovery, including anxiety symptoms (Figure 5). OM also increased AMPK activity and reduced oxidative stress-induced senescence (Figure 6B–E). This demonstrates that OM has neuroprotective properties [60]. An interesting aspect of the results is that OM reduced the expression of RAGE in the D-Gal senescence model (Figure 6A). Excessive D-Gal accumulation forms AGEs through non-enzymatic glycosylation. They then bind to RAGE, which activates inflammatory pathways and increases oxidative stress [61]. Furthermore, bounding with RAGE occurs with NADPH oxidase, thus increasing oxidative stress [62]. Additionally, AGEs can promote the aggregation of Amyloidβ (Aβ). A series of processes involving AGEs may be a major cause of Alzheimer’s disease [63]. Therefore, suppressing these processes by using OM could be a useful strategy in the treatment of Alzheimer’s disease. In the case of heart tissue, we observed that OM restored the activity of AMPK and increased the activity of autophagy (Figure 7A–F). OM also showed a reduction in oxidative stress-induced senescence (Figure 7G–I). AMPK is the center of cellular energy metabolism. This maintains energy homeostasis and activates autophagy to recycle energy [22,23]. Autophagy occurs through the phosphorylation of ULK1 (Unc-51-like kinase 1) and the inhibition of mTOR [64]. Upon investigating the cardiac tissue, it was evident that OM activates the AMPK pathway, therefore reducing oxidative stress-induced damage and senescence through autophagy. This maintains cell survival and homeostasis.

Consistent with the results of previous studies, this study did not find out what operating principle exists between Oxymatrine and AMPK and other pathways. Regarding future studies, it is necessary to conduct experiments by linking the fact that OM is a quinolizidine ring structure to the molecular pathway. Additionally, in this study, we did not examine how the downstream pathways function regarding AMPK and senescence. Based on the claimed schematic diagram, further research is needed on the antioxidant pathway and autophagy. In particular, there is a need to focus on SIRT1, since it is highly related to senescence.

Non-coding RNAs, especially long non-coding RNAs (lncRNAs), have a massive impact on several disease mechanisms. As mentioned above, OM has antioxidant effects by activating the SIRT1 pathway. Several studies have shown that long non-coding RNAs have the tendency to interact with SIRT1 in several diseases/conditions, including cardiac failure, non-alcoholic fatty liver disease, and ischemic stroke, most of which are considered as aging-related diseases [65]. Our data showed the positive effect of OM treatment when used against cardiac and brain tissues in the oxidative stress-induced senescence mice model. These positive effects could be due to the interaction between lncRNA and SIRT1, activated by OM.

Though the above study shows a positive interaction between lncRNAs, SIRT1, and OM, a recent study on centenarians showed that dysregulated lncRNAs are the possible preventive factors in healthy aging in the longevous population [66]. This could be due to the fact that lncRNAs have been observed to mediate oxidative stress in several disorders [67]. The silencing of several lncRNAs has been shown in the recovery of neuronal activities. These contradicting phenomena have been mentioned in many studies. The role of some of lncRNAs, either positive or negative, is tissue-specific. One example is the non-coding RNA MALAT1, which, during endothelial injury, played a positive role by acting against oxidative stress, whereas it elevates oxidative stress in epithelial cells. OM-resistant cancer cells showed a high level of MALAT1 expression, and this was associated with poor survival [68]. This shows that active OM treatment negatively regulates MALAT1 and reduces oxidative stress.

This study has some limitations. This study did not show how AMPK and its downstream regulators are activated upon treatment with OM, nor how OM interacts with SIRT1, AMPK, and lncRNAs and modulates antioxidant effects. These biochemical interactions should be studied and explained in further studies.

In summary, OM ameliorated senescence by AMPK activation in oxidative stress-induced senescence cells and animal models. We suggest that OM could be a useful tool in treatment strategies for neurodegenerative diseases by reducing RAGE. We have also found that mediating autophagy could regulate tissue homeostasis. Therefore, we suggest that the interaction between AMPK activation and oxidative stress has an important role in anti-senescence. As in previous studies, OM itself can have antioxidant effects. In this study, we focused on the AMPK pathway. As a result, OM restored AMPK activity and ameliorated senescence. As shown in the schematic diagram in Figure 8, OM treatment enhances the activity of AMPK, which directly stimulates the Nrf2 pathway, resulting in an increase in antioxidants. This also leads to a reduction in oxidative stress. In addition, NAD+ (nicotinamide adenine dinucleotide) increases and interacts with sirt1. This interaction activates autophagy to reduce cellular damage and enhance cell and tissue homeostasis. All these effects result in a reduction in cellular senescence. Ultimately, the interrelationship between AMPK and oxidative stress indicates that it has important functions in homeostasis, senescence, and metabolic disorders. In a follow up study, we plan to reaffirm the sirt1/AMPK pathway and further extend it to find out whether the Nrf2 pathway shows an increase in antioxidants to evaluate the accuracy of the schematic diagram.

## Figures and Tables

**Figure 1 antioxidants-12-02078-f001:**
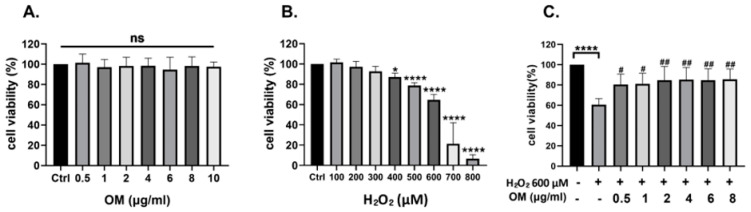
The protective effect of Oxymatrine against H_2_O_2_ in HT22 cells. (**A**) The toxicity of Oxymatrine was determined by conducting the MTT assay. (**B**) The toxicity of H_2_O_2_ was observed as the concentration increased. (**C**) The protective effect of OM against H_2_O_2_ was checked. All data are represented as the mean ± standard deviation (SD) (*n* = 3). *^/#^ *p* < 0.05, ^##^ *p* < 0.01, and **** *p* < 0.0001. # is the statistical significance between the senescence model and the OM group.

**Figure 2 antioxidants-12-02078-f002:**
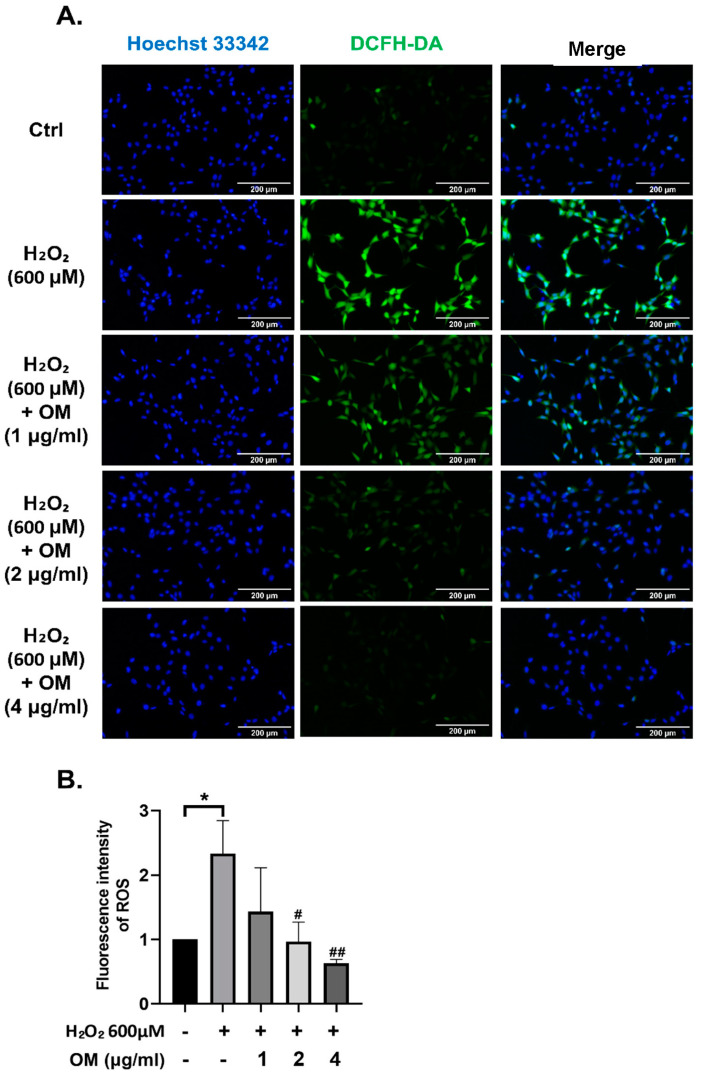
Reduction in ROS by Oxymatrine in the HT22 cells. (**A**) The effect of OM on ROS was determined by the DCFH-DA assay. (**B**) Bar plot showing the quantified fluorescence. ROS decreased depending on the concentration of OM. All data are represented as the mean ± standard deviation (SD) (*n* = 3). *^/#^ *p* < 0.05 and ^##^ *p* < 0.01. # is the statistical significance between the senescence model and the OM group. Scale bar = 200 μm.

**Figure 3 antioxidants-12-02078-f003:**
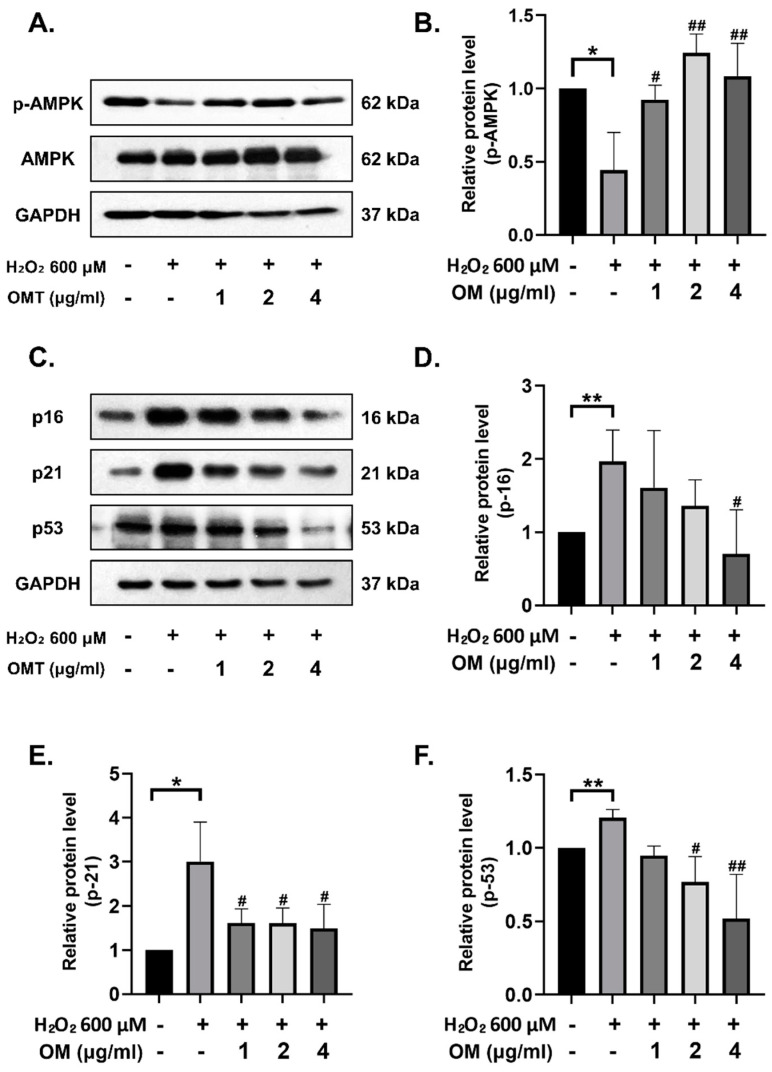
Oxymatrine reduces senescence via AMPK activity in HT22 cells. (**A**) Representative images of Western blot showing AMPK and phosphorylated AMPK expression levels. (**B**) The activity of AMPK is restored. The quantification of the blots was performed using ImageJ software. (**C**) Representative images of immunoblot against senescence markers p53, p21, and p16. (**D**–**F**) p53, p21, and p16 were all decreased. The expression levels of the senescence markers were quantified using ImageJ software. All data are represented as the mean ± standard deviation (SD) (*n* = 3). *^/#^ *p* < 0.05 and **^/##^ *p* < 0.01. # is the statistical significance between the senescence model and the OM group.

**Figure 4 antioxidants-12-02078-f004:**
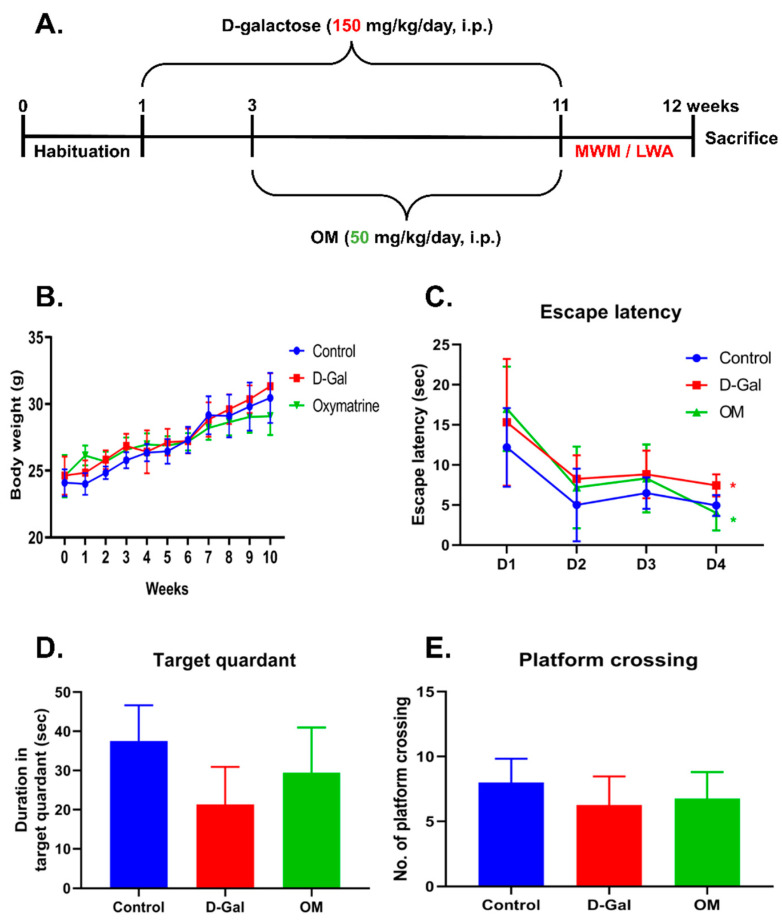
Oxymatrine improves memory loss in D-Gal-induced senescence mice. (**A**) Schematic diagram showing the experimental plan. (**B**) Body weights of the mice. (**C**–**E**) Behavioral test (Morris water maze test). Each result represents the platform discovery time, dwell time, and crossing result in the target quadrant. (**C**) On the fourth day, the discovery time of the control and OM groups was shorter than that of the D-gal group. (**D**) Platform dwell time was lengthened by OM administration. (**E**) The number of crossings slightly increased. All data are represented as the mean ± standard deviation (SD) (*n* = 4). * *p* < 0.05.

**Figure 5 antioxidants-12-02078-f005:**
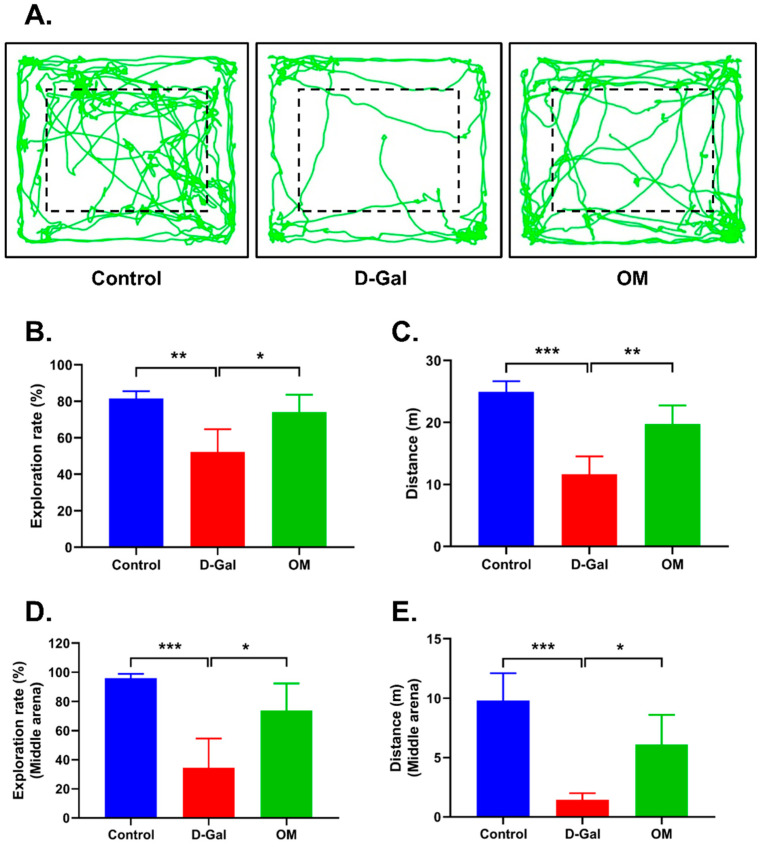
Oxymatrine alleviated anxiety symptoms in D-Gal-induced senescence mice. (**A**) Behavioral test based on locomotor activity in an open field test. (**B**,**C**) The exploration and distance to the whole and (**D**,**E**) the exploration and distance to the center. All data are represented as the mean ± standard deviation (SD) (*n* = 4). * *p* < 0.05, ** *p* < 0.01, and *** *p* < 0.001.

**Figure 6 antioxidants-12-02078-f006:**
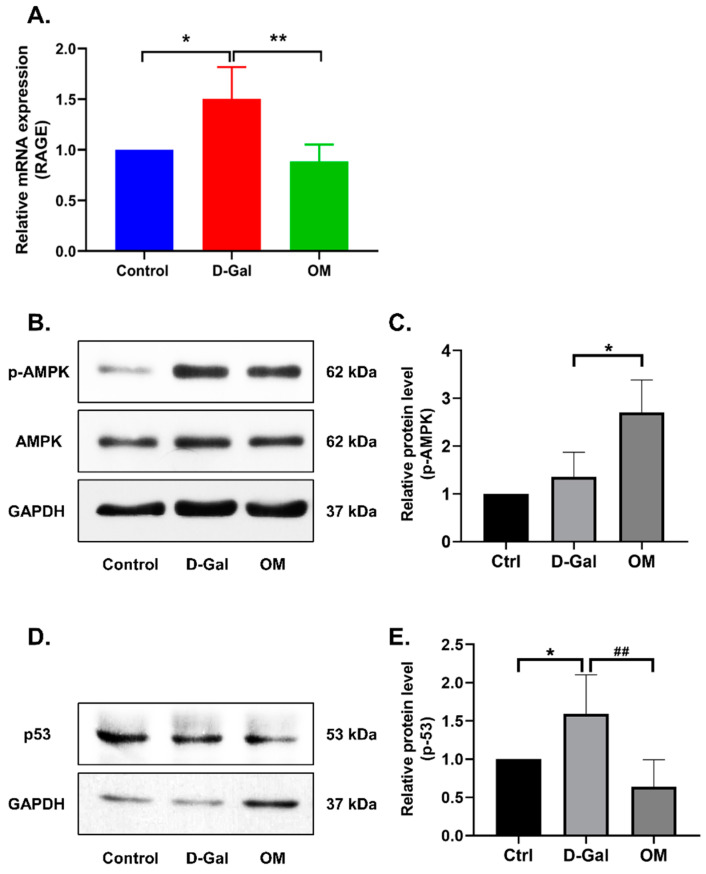
The ameliorative effect of Oxymatrine in D-Gal-induced senescence mice hippocampal tissue. (**A**) Bar plot showing the relative fold changes in the expression of RAGE mRNA. (**B**) The representative images of the immunoblot against AMPK and p-AMPK from the hippocampus. (**C**) The activity of AMPK was restored. (**D**) Representative images of the immunoblot against p53 and GAPDH from the hippocampus. (**E**) p53 was observed to decrease. Protein expression was quantified using ImageJ software v1.4.3.x. All data are represented as the mean ± standard deviation (SD) (*n* = 3–4). * *p* < 0.05 and **^/##^ *p* < 0.01. # is the statistical significance between the senescence model and the OM group.

**Figure 7 antioxidants-12-02078-f007:**
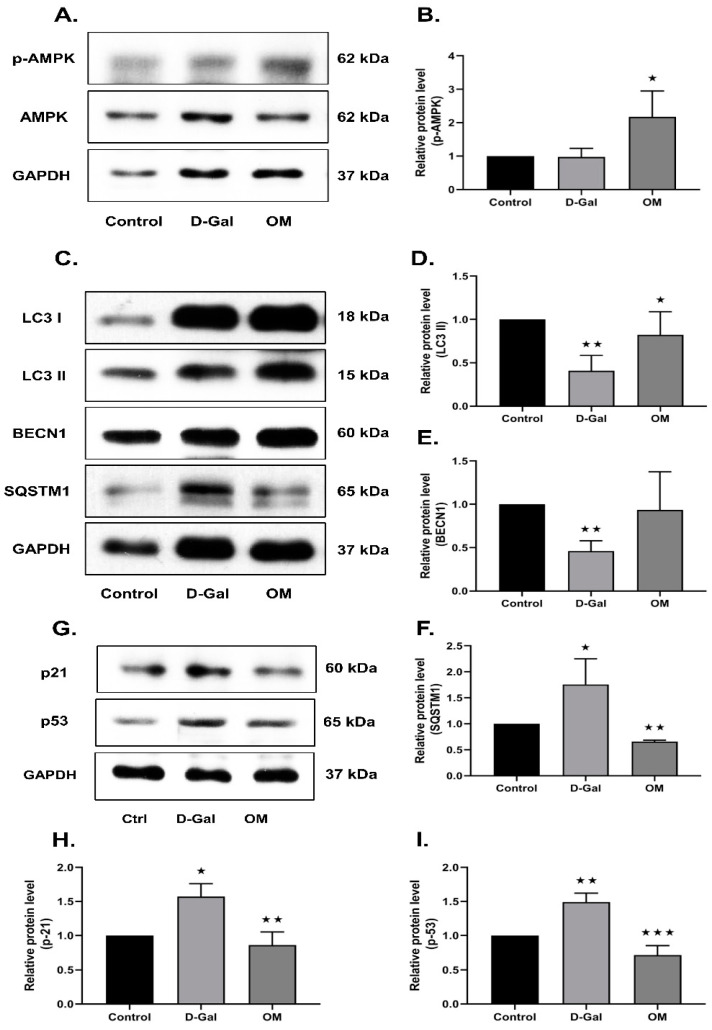
Ameliorative effect of Oxymatrine on D-Gal-induced senescence mice cardiac tissue. (**A**,**B**) Representative images of the immunoblot against AMPK and p-AMPK from heart. The activity of AMPK was restored. (**C**–**F**) Representative images of immunoblot against LC3 I, LC3 II, BECN1, SQSTM1, and GAPDH from the heart tissue. Autophagy markers were restored. (**G**–**I**) Representative images of the immunoblot against p53, p21, and GAPDH from the heart tissue. p53 and p21 can be observed to decrease. (**B**–**I**) Protein expression was quantified using ImageJ software. All data are represented as the mean ± standard deviation (SD) (*n* = 3–4). * *p* < 0.05, ** *p* < 0.01, and *** *p* < 0.001.

**Figure 8 antioxidants-12-02078-f008:**
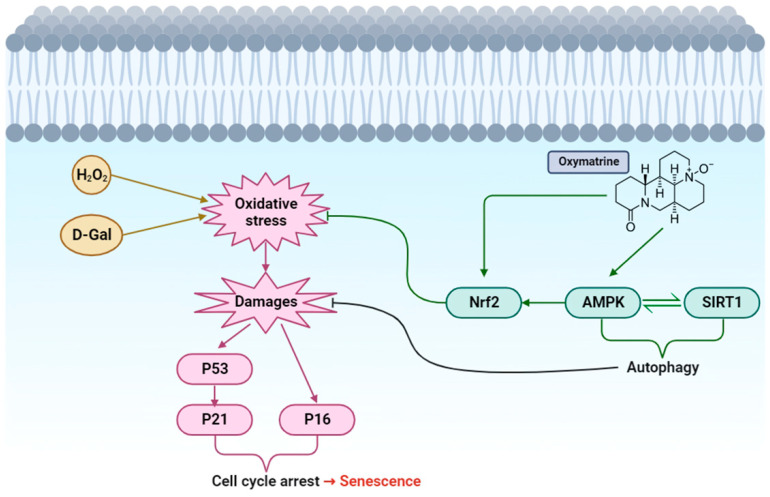
Overall schematic representation of how Oxymatrine responds to oxidative stress and senescence. Oxymatrine causes the activation of AMPK. AMPK activates the antioxidant pathway. Additionally, AMPK interacts with SIRT1 to mediate autophagy. Reduced oxidative stress and cell damage prevents senescence.

## Data Availability

The data presented in this study are available from the corresponding author upon request.

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
