# Peer review of "Oxymatrine Improves Oxidative Stress-Induced Senescence in HT22 Cells and Mice via the Activation of AMP-Activated Protein Kinase"

_antioxidants, 2023, doi:10.3390/antiox12122078_

Round 1

Reviewer 1 Report

Comments and Suggestions for Authors

The report by Maharajan et al. is of interest but lacks explanation/clarification.

1.Title needs to be changed to include in vitro cell and in vivo mouse information

2.Abstract lacks description of the concentrations of OM for experiments performed.

3. It is not clear how many animals were used in the stated experiments and what statistical analysis was performed, especially when there were only 4 mice per group.

4. How was the IP dosing of 50 mg/Kg determined? See lines 158 and 262.

5. What was the diet of the mice?  Did the diet contain phytoestrogens, if so at what concentration or level PPM?  If phytoestrogens were present in the diet what are the interactions of OM with these antioxidant compounds?

6. How did the authors control for the 10 % FBS in the cell cultures, line 12 since this contains steroids?

7. Figures 1, 2, 3 and 6 the symbol # has no description or explanation.

8. The discussion only has 10 references, and for the most part is a repeat of the results, please discuss the results using previous reported studies on OM.

9. The authors overstate that OM might be a treatment for Alzheimer's disease when the dosing was 50 mg/Kg, please re-write. Why is the dosing effective in vitro at the microgram level but is only effective at 50 mg/Kg in vitro?

10. Please state in the discussion the limitations of the experiments/studies.

11.  The true significance of this report is unknown based upon the poor presentation of the studies.

Minor:

Lines 43, 83 and 85 change to: in vitro and in vivo

Comments on the Quality of English Language

Suggest minor English correction

Author Response

Q 1. Title needs to be changed to include in vitro cell and in vivo mouse information.

Ans. We were changed the title to ‘Oxymatrine improves the oxidative stress-induced senescence in HT22 cells and mice via activation of AMPK.’.

Please confirm.

Q 2. Abstract lacks description of the concentrations of OM for experiments performed.

Ans. We additionally described the concentration of oxymatrine in the Abstract section.

‘By treating 1, 2 and 4 μg/ml of OM to the HT22 mouse cell line and by administering 50 mg/kg OM to mice we evaluated improvement effect.’ (line 17 and 18)

Q 3. It is not clear how many animals were used in the stated experiments and what statistical analysis was performed, especially when there were only 4 mice per group.

Ans. For the animal study we used n=3-4 and Ordinary one-way or two-way ANOVA multiple comparisons.

Fig: 4B. n=4; Ordinary one-way ANOVA (Tukey's multiple comparisons test)

Fig: 4C. n=4; Ordinary two-way ANOVA (Tukey's multiple comparisons test)

Fig: 4D-E. n=4; Ordinary one-way ANOVA (Tukey's multiple comparisons test)

Fig: 5B-E. n=4; Ordinary one-way ANOVA (Sidak's multiple comparisons test)

Fig: 6A-E. n=4; Ordinary one-way ANOVA (Tukey's multiple comparisons test)

Fig: 7A-I. n=3-4; Ordinary one-way ANOVA (Tukey's multiple comparisons test)

Q 4. How was the IP dosing of 50 mg/Kg determined? See lines 158 and 262.

Ans. In this study we have focused on the oxidative stress-induced senescence in the heart and hippocampus tissues. For that we have selected the oxymatrine concentration based the literature that is “medium dose of 50 mg/Kg oxymatrine for long term (8 weeks) IP administration”.

Ref:

  1. Oxymatrine protects against the effects of cardiopulmonary resuscitation via modulation of the TGF-β1/Smad3 signaling pathway
  2. Protective effect of oxymatrine on chronic rat heart failure
  3. Oxymatrine Ameliorates Doxorubicin-Induced Cardiotoxicity in Rats

Q 5. What was the diet of the mice?  Did the diet contain phytoestrogens, if so at what concentration or level PPM?  If phytoestrogens were present in the diet what are the interactions of OM with these antioxidant compounds?

Ans. In this study all animals were fed normal chow pellets (ND, SAM #31, Samtako Inc.).

(http://www.samtako.com/bbs/board.php?bo_table=b2_2_2)

Q 6. How did the authors control for the 10 % FBS in the cell cultures, line 12 since this contains steroids?

Ans. Steroids, including hormones like estrogen and progesterone, are naturally occurring compounds in the blood of animals, including fetal bovines. FBS may contain trace amounts of these steroids, but the levels are typically very low and are not usually a concern in cell culture applications.

Therefore, we believe that steroids are not a problem in cell culture.

Q 7. Figures 1, 2, 3 and 6 the symbol # has no description or explanation.

Ans. We have supplemented the explanation of the symbol #.

‘(*/#p <0.05, **/##p <0.01, ***/###p < 0.001, ****/#####p < 0.0001).’ and ‘# is the statistical significance between the senescence model and the OM group.’ (line 202 and 203 / Figure legend 1, 2, 3 and 6)

Q 8. The discussion only has 10 references, and for the most part is a repeat of the results, please discuss the results using previous reported studies on OM.

Ans. We supplemented the discussion section based on existing research.

Sirt1 interacts with AMPK [45, 46]. This means that OM can activate AMPK.AMPK maintains homeostasis of energy metabolism. So, AMPK is called a metabolic master switch. Additionally, AMPK confers adaptive properties to stress [47]. This role of AMPK is associated with senescence [48]. From a metabolic perspective, excessive energy consumption accelerates aging [49]. Supporting this, there is a report that calorie restriction slows down aging [50-52]. Therefore, maintaining AMPK activity is closely related to energy homeostasis and longevity [23]. Oxidative stress is closely related to senescence [18]. AMPK enhances oxidative stress resistance through its downstream regulators [53]. there is an increase in antioxidants via the Nrf2 pathway [54]. In another pathway, AMPK interacts with Sirt1 to regulate cellular energy metabolism, apoptosis, cell proliferation, and inflammatory responses. And Sirt1 directly activates autophagy [55]. Ultimately, AMPK regulates metabolic homeostasis and cell survival during metabolic and oxidative stress [56]. And AMPK regulates aging-related transcription factors [57]. Activation of AMPK has also been shown to alleviate ischemia/reperfusion symptoms and have neuroprotective effects [58, 59]. These functions are associated with the biochemical characteristics of OM. this suggests that OM can activate AMPK. As such, OM is closely related to AMPK and is thought to be able to respond to oxidative stress. This is thought to reduce cellular senescence. (line 346 - 364)

Q 9. The authors overstate that OM might be a treatment for Alzheimer's disease when the dosing was 50 mg/Kg, please re-write. Why is the dosing effective in vitro at the microgram level but is only effective at 50 mg/Kg in vitro?

Ans. We changed the phrase 'potential therapies' to 'useful strategy'.

‘Therefore, suppressing these processes by OM could be a useful strategy in Alzheimer's disease.’ (line 379 and 380) and We suggest that OM could be a useful strategy for neurodegenerative diseases by reducing RAGE. (line 397 and 398)

Q 10. Please state in the discussion the limitations of the experiments/studies.

Ans. We have mentioned the limitations of the current study and are planning future plans.

‘Along with previous research results, this study did not find out what operating principle exists between Oxymatrine and AMPK and other pathways. In the future, I think it is necessary to conduct experiments by linking the fact that OM is a quinoliz-idine ring structure to the molecular pathway. Additionally, in this study, we do not know how the downstream pathways function between AMPK and senescence. Based on the claimed schematic diagram, further research is needed on the antioxidant pathway and autophagy. In particular, I think there is a need to focus on Sirt1 since it is highly related to senescence.’ (line 389 - 395)

Q 11. The true significance of this report is unknown based upon the poor presentation of the studies.

Ans. The ultimate goal of our research is to find substances that serve as strategies to reduce senescence. Among the causes of cellular senescence, regulation of oxidative stress is very important. Excessive stress can disrupt cellular homeostasis. The pathway that links oxidative stress and cellular homeostasis is AMPK. AMPK stabilizes metabolism through energy regulation and maintains homeostasis. Additionally, AMPK maintains homeostasis through stress regulation. AMPK's response to all these stresses is associated with aging and ultimately longevity. We chose Oxymatrine to check the response of AMPK. There is a reason why AMPK and Oxymatrine were linked. First, there are reports of activating the Nrf2 pathway. It is associated with antioxidants. Second, it activates the Sirt1 pathway. Sirt1 activity is important in senescence. Sirt1 also interacts with AMPK. Lastly, both AMPK and Oxymatrine have ischemia/reperfusion symptom relief and neuroprotective effects. Looking at these functions of AMPK and the biochemical characteristics of Oxymatrine, it was thought that Oxymatrine could activate AMPK. As a result, the results of this study showed that Oxymatrine actually acted on activating AMPK. Additionally, it was confirmed that senescence is reduced.

Ref:

  1. Oxymatrine ameliorates renal ischemia-reperfusion injury from oxidative stress through Nrf2/HO-1 pathway.
  2. Activation of Nrf2/HO-1 signaling: An important molecular mechanism of herbal medicine in the treatment of atherosclerosis via the protection of vascular endothelial cells from oxidative stress
  3. Oxymatrine attenuates oxidized low‑density lipoprotein‑induced HUVEC injury by inhibiting NLRP3 inflammasome‑mediated pyroptosis via the activation of the SIRT1/Nrf2 signaling pathway
  4. Oxymatrine attenuates cognitive deficits through SIRT1-mediated autophagy in ischemic stroke
  5. Neuroprotective Effects of Oxymatrine via Triggering Autophagy and Inhibiting Apoptosis Following Spinal Cord Injury in Rats

6. AMP-activated protein kinase (AMPK) controls the aging process via an integrated signaling network

Reviewer 2 Report

Comments and Suggestions for Authors

In the present study, the anti-senescence potential of the alkaloid oxymatrine through oxidative stress-induced in vitro and in vivo models was evaluated. By treating 600 μM of H2O2 to the HT22 mouse hippocampal neuronal cell line and by administering 150 mg/kg D-galactose to mice was generated the oxidative stress-induced senescence models. We evaluated different senescence markers, AMPK activity, and autophagy along with DCFH-DA detection reaction and behavioral tests. In HT22 cells, OM showed a protective effect. OM by reducing ROS and increasing p-AMPK expression could potentially reduce oxidative stress-induced senescence. In the D-Gal-induced senescence mouse model, both the brain and heart tissues recovered AMPK activity, resulting in reduced levels of senescence. In neural tissue, to assess neurological recovery, including anxiety symptoms and exploration, was used the behavioral Test. Along with it was found that OM decreased the expression level of receptors for advanced glycation end products (RAGE). In heart tissue, restoration of AMPK activity also increased the activity of autophagy. Therefore, the obtained results suggested that OM ameliorates oxidative stress-induced senescence through its antioxidant action by restoring AMPK activity. In my opinion, this work was well conducted and displayed interesting results. Therefore, I suggest its acceptance after minor revision, as follows:

1. Please, indicate if OM was isolated from a natural source or if it was purchased from a chemical company.

2. Details concerning the purity and chemical characterization of the tested compound must be included in the manuscript.

3. Some minor spelling mistakes must be corrected in the text – additionally, avoid the use of “we” in the manuscript.

4. Biological assays were adequately conducted but no information concerning the approval of these experiments by the Ethical committee was provided in the manuscript. This point must be well explained.

Author Response

Q 1. Please, indicate if OM was isolated from a natural source or if it was purchased from a chemical company.

Ans. We purchased Oxymatrine (#CFN99805; purity ≥ 99.4%) from a company called ChemFaces (430056; Wuhan, Hubei).

‘ChemFaces (430056; Wuhan, Hubei) supplied Oxymatrine (#CFN99805; purity ≥ 99.4%).’ (line 94 and 95).

Q 2. Details concerning the purity and chemical characterization of the tested compound must be included in the manuscript.

Ans. We explained purity (purity ≥ 99.4%) in the Materials and Methods section. And in the introduction, we explained the biochemical properties of Oxymatrine.

‘Oxymatrine (OM) is a natural quinolizidine alkaloid compound extracted from the roots of Sophora flavescens. Molecular Formula is C15H24N2O2. It is commonly known for its hepatoprotective role and has antioxidant effects because it has a quinolizidine ring structure [30-32]. Recent studies have revealed that OM has various functions, including antibacterial, antiviral, anti-inflammatory, and anticancer [33, 34]. Various studies have also shown its role in improving blood pressure regulation and ischemia/reperfusion injury in the heart, and has a protective effect on nerves, so it has been used in neurodegeneration models [35-37].’ (line 72 - 79) and ‘ChemFaces (430056; Wuhan, Hubei) supplied Oxymatrine (#CFN99805; purity ≥ 99.4%).’ (line 94 and 95).

Q 3. Some minor spelling mistakes must be corrected in the text – additionally, avoid the use of “we” in the manuscript.

Ans. We have corrected the grammatical issues mentioned.

Q 4. Biological assays were adequately conducted but no information concerning the approval of these experiments by the Ethical committee was provided in the manuscript. This point must be well explained.

Ans. We follow the guidelines of the Declaration of Helsinki, and animal experiments were approved by the ethics committee of Chosun University. The approval number is (CIACUC2020-A0009, 31 March 2020).

It was written in the Materials and Methods section of the main text and the lower part of the manuscript.

‘The animal Care and Use committee, of Chosun University have sanctioned all the animal experiments (CIACUC2020-A0009).’ (line 153 -155) and ‘Institutional Review Board Statement: The study was conducted according to the guidelines of the Declaration of Helsinki and approved by Ethics Committee of Chosun University (CIACUC2020-A0009, 31 March 2020).’ (line 425 - 427)

Round 2

Reviewer 1 Report

Comments and Suggestions for Authors

The responses by the authors have not addressed points 4, 5, 6 and 9 sufficiently. Especially, how 50 mg/Kg might be an effective dose in health applications. There has been a small improvement of this manuscript in the revised version but it has not been sufficiently improved to warrant publication.

Comments on the Quality of English Language

needs to be reviewed by English department of the journal

Author Response

Q 4. How was the IP dosing of 50 mg/Kg determined? See lines 158 and 262.

Ans. As we have mentioned in the Cell viability Assay under the subtitle materials and methods the concentration of OM was selected by dose dependent treatment of OM on the cells ranging from 0.5 to 10 μg/ml and we have not seen a drastic toxic effects in the cells. Therefore, we have used 3 different doses in the experiments. In this study we did a literature review and used 50mg/kg, as most of the studies showed this as the best concentration in protecting various model organisms. The referred journals are mentioned below.

When the concentration was converted to µM, in the in vitro experiments we used 3.78µM, 7.56 µM and 15.12 µM. Similarly in the in vivo experiments the mice weight throughout the experiment was ranging from 25gms to 30 gms approximately. As per the journal review, we chose 50mg/Kg as its not toxic to the animals and this translates in to 4.725 µM to 5.67 µM. This is closer to the safe concentrations that we have used for our in vitro studies too.

Ref:

  1. Wang D., et al., Oxymatrine protects against the effects of cardiopulmonary resuscitation via modulation of the TGF-β1/Smad3 signalling pathway. Mol Med Rep, 2018. 17(3): p. 4747-4752:
  2. Hu ST., et al., Protective effect of oxymatrine on chronic rat heart failure. J Physiol Sci, 2011; 61(5): p.363-72.
  3. Zhang Y, Yi M, Huang Y. Oxymatrine Ameliorates Doxorubicin-Induced Cardiotoxicity in Rats. Cell Physiol Biochem, 2017; 43(2): p.626-635.

Q 5. What was the diet of the mice?  Did the diet contain phytoestrogens, if so at what concentration or level PPM?  If phytoestrogens were present in the diet what are the interactions of OM with these antioxidant compounds?

Ans. In this study all animals were fed normal chow pellets (ND, SAM #31, Samtako Inc.).

(http://www.samtako.com/bbs/board.php?bo_table=b2_2_2). According to the manufacturers this is a standard diet and only the special diets (high fat diets) contain the soybean oil, which is a source of phytoestrogens. The diet of the mice in this study only consists of Protein, fat, calcium, phosphorus along with other essentials like carbohydrates. Therefore, in the diet of the mice phytoestrogens are absent. We have given the link for the contents of the mice feed provided by the suppliers and, we are attaching a pdf file of it for the reviewer’s concern.

Q 6. How did the authors control for the 10 % FBS in the cell cultures, line 12 since this contains steroids?

Ans. In the study we cultured the cells with 10%FBS prior to any experiments. During subculture and seeding the cells for further experiments we have used serum free media in all the conditions including control. Once the cells are settled down after 24 hours from the seeding time in the serum free media, we treat the compounds. Therefore, we need not to be worried about the contents of the FBS. We have missed this to mention in the previous version of the manuscript. We have updated this point in the current version of the manuscript.

Q 9. The authors overstate that OM might be a treatment for Alzheimer's disease when the dosing was 50 mg/Kg, please re-write. Why is the dosing effective in vitro at the microgram level but is only effective at 50 mg/Kg in vitro?

Ans. As in the previous version of the revision we changed the phrase 'potential therapies' to 'useful strategy' in the following sentences in the manuscript.

‘Therefore, suppressing these processes by OM could be a useful strategy in Alzheimer's disease.’ (line 379 and 380) and We suggest that OM could be a useful strategy for neurodegenerative diseases by reducing RAGE. (line 397 and 398).

Also, we have observed the protective effect of OM in 50mg/kg concentration in the brain tissues, we emphasize this as the best dosage for Alzheimer’s disease.

The dosing was effective in vitro at the microgram level and it is only effective in the animal at mg/kg concentration is due to the body surface area scaling. According to this, the administration of a drug and absorption of it in the blood depends on numerous factors. In general, oxygen utilization, basal metabolism, circulating plasma protein, caloric expenditure, and blood volume. Similarly, safety factors should also be considered when deciding the dose. Therefore, we chose 50mg/kg as the literature showed that as a middle ground concentration. Also, as we mentioned in the answer for the 4th comment the molar concentration in both in vitro and in vivo experiments are closer to each other’s. Therefore, we could deduce that the dosing is effective in both in vitro and in vivo, though one is in microgram level and the other in milligram level.

Ref:

1) Shin J, Seol I, Son C, Interpretation of Animal Dose and Human Equivalent Dose for Drug Development, J Korean Med. 2010; 31(3):0

2) Blanchard O, Smoliga J, Translating dosages from animal model to human clinical trials-revisiting body surface area scaling. FASEB J.2015. 29(5):1629-34.

As the reviewer suggested us to check for some minor English revision we have also thoroughly checked for English language and made some modifications in the final manuscript.

Reviewer 2 Report

Comments and Suggestions for Authors

The authors adequately answered all points I asked in my previous analysis of this paper. Therefore, I suggest its acceptance in the revised form.

Author Response

NA